# Selection of stimulus parameters for enhancing slow wave sleep events with a neural-field theory thalamocortical model

**Felipe A. Torres**[1,3☉], **Patricio Orio**[2,3☉]*, **María-José Escobar**[1☉]*

**1** Department of Electronic Engineering, Universidad Técnica Federico Santa María, Valparaíso, Chile,
**2** Centro Interdisciplinario de Neurociencia de Valparaíso, Universidad de Valparaíso, Valparaíso, Chile,
**3** Advanced Center for Electrical and Electronic Engineering (AC3E), Valparaíso, Chile

☉ These authors contributed equally to this work.
\* patricio.orio@uv.cl (PO); mariajose.escobar@usm.cl (M-JE)

**Data Availability Statement:** The data/code is available at: https://github.com/mjescobar/SWSNeuralField.

## Abstract

Slow-wave sleep cortical brain activity, conformed by slow-oscillations and sleep spindles, plays a key role in memory consolidation. The increase of the power of the slow-wave events, obtained by auditory sensory stimulation, positively correlates with memory consolidation performance. However, little is known about the experimental protocol maximizing this effect, which could be induced by the power of slow-oscillation, the number of sleep spindles, or the timing of both events' co-occurrence. Using a mean-field model of thalamocortical activity, we studied the effect of several stimulation protocols, varying the pulse shape, duration, amplitude, and frequency, as well as a target-phase using a closed-loop approach. We evaluated the effect of these parameters on slow-oscillations (SO) and sleep-spindles (SP), considering: (i) the power at the frequency bands of interest, (ii) the number of SO and SP, (iii) co-occurrences between SO and SP, and (iv) synchronization of SP with the up-peak of the SO. The first three targets are maximized using a decreasing ramp pulse with a pulse duration of 50 ms. Also, we observed a reduction in the number of SO when increasing the stimulus energy by rising its amplitude. To assess the target-phase parameter, we applied closed-loop stimulation at 0°, 45°, and 90° of the phase of the narrow-band filtered ongoing activity, at 0.85 Hz as central frequency. The 0° stimulation produces better results in the power and number of SO and SP than the rhythmic or random stimulation. On the other hand, stimulating at 45° or 90° change the timing distribution of spindles centers but with fewer co-occurrences than rhythmic and 0° phase. Finally, we propose the application of closed-loop stimulation at the rising zero-cross point using pulses with a decreasing ramp shape and 50 ms of duration for future experimental work.

## Author summary

During the non-REM (NREM) phase of sleep, events that are known as slow oscillations (SO) and spindles (SP) can be detected by EEG. These events have been associated with the consolidation of declarative memories and learning. Thus, there is an ongoing interest

**Funding:** This work was supported by ANID-Basal Project FB0008 (P.O. www.anid.cl), ANID grant PhD scholarship No. 2118640 (F.A.T. www.anid.cl), AFOSR Grant Nro. FA9550-19-1-0002 (M.-J.E. https://www.afrl.af.mil), and ANID-Millennium Science Institute ICN09-022 (P.O. www.anid.cl). The funders had no role in study design, data collection and analysis, decision to publish, or preparation of the manuscript.

**Competing interests:** The authors have declared that no competing interests exist.

in promoting them during sleep by non-invasive manipulations such as sensory stimulation. In this paper, we used a computational model of brain activity that generates SO and SP, to investigate which type of sensory stimulus –shape, amplitude, duration, periodicity– would be optimal for increasing the events' frequency and their co-occurrence. We found that a decreasing ramp of 50 ms duration is the most effective. The effectiveness increases when the stimulus pulse is delivered in a closed-loop configuration triggering the pulse at a target phase of the ongoing SO activity. A desirable secondary effect is to promote SPs at the rising phase of the SO oscillation.

## Introduction

Humans spend about one-third time of their life sleeping. This behavior has paramount importance for the process of learning, as it contributes to the consolidation of memories [1, 2]. During the non-rapid eye movement (NREM) phase of sleep, the brain's electrical activity is characterized by the occurrence of events recognizable on the cortical activity measured by electroencephalogram (EEG) registers. These events are the slow oscillations (SO), single high-amplitude cortical oscillations lasting 0.8 to 2 seconds [3]; and the sleep spindles (SP), thalamo-cortical oscillation bursts lasting 0.5 to 2 seconds in the 9-16 Hz frequency band [4]. NREM is also recognizable by events on the hippocampus activity, the sharp wave-ripples, oscillations bursts in the 100-250 Hz frequency range of 50–100 ms duration that can be measured with invasive intracranial electrodes [5].

The EEG measures electrical potentials from the cortex on the scalp. The characteristics of the EEG waveform depend on the contribution from multiple electrical current sources [6]. Characteristics such as the amplitude and frequency also involve the interaction of the cortex with other cerebral structures. It is well known that sleep and arousal rhythms, including SOs and SPs, are generated in the thalamus and cortex [7, 8]. Moreover, similar to in-vivo activity can be reproduced by computer models of the corticothalamic system [9, 10].

On the other hand, memory consolidation is a complex problem where different mechanisms play a role. The temporal coincidence of sharp wave-ripples, the sleep spindles, and the slow oscillations promote the transformation of short-term memories stored in hippocampus connections to long-term storage memories by increasing the synaptic strength of cortical neural networks. This perspective is known as the Standard Consolidation Theory [11], and only occurs during the deepest NREM sleep stage [5, 12]. On the other hand, REM sleep theta-gamma oscillations coupling, and the alternation between REM and NREM stages also impact memory consolidation [13].

Regarding slow wave sleep (SWS), NREM stage of slow frequency activity with SOs and SPs' presence, the literature points out that the number of spindles appearances locked to the UP phase of slow oscillations could play a more significant role in cortical memory formation [14–16]. This has led to a standing interest in improving the consolidation process –and thus long-term memory– by non-invasive sensory stimulation or direct electromagnetic stimulation while the brain is in SWS [17, 18].

The physiological goal of stimulation during SWS is to increase the power and the occurrences of both events (SOs and SPs), including their precise timing coincidence. An increase of sharp-wave ripples is also desirable [19], but there is no way to measure hippocampus activity with non-invasive methods. Further, stimulation for memory consolidation improvement should not disturb sleep cycles [2]. Stimulating during SWS also takes advantage of a decreased awareness to external stimuli during this sleep stage.

There is evidence supporting that sensory manipulation of SO leads to memory improvement [3] or disruption [20], with significant effects in enhancing the declarative memory and no impact on procedural memory [20, 21]. Moreover, the number of SPs during SWS, and their power, is also related with best results in word-pairs memory tests [2]. However, the knowledge and optimization of stimulation protocols have been based on correlation analysis rather than causal evidence [22].

Even if auditory closed-loop stimulation has been used to improve the memory consolidation process, there is no evidence of tuning the stimulation protocol to maximize this effect. Specifically, sensory stimulation looks to maximize the correct timing between the evoked SP and SO events associated with the memory consolidation process. The manipulation of SP in auditory closed-loop [23], and the use of memory cues (sounds related to the learning episode), show better results when the external stimulus is applied at a specific phase of the SO [24].

An alternative approach to improving the stimulation paradigm that attempts to increase SP and SO events during SWS is the use of mathematical models of brain activity [25, 26]. In this way, mechanistic insights about the generation of SP and SO activity can guide the search for a better stimulation that maximizes its efficacy. In this work, we present a systematic study of possible stimulation protocols and the more efficient features of the stimuli in promoting these SWS events' co-occurrence.

Specifically, in this work we use a model based on the Neural Field Theory (NFT) [27, 28] with corticothalamic loops between two cortical neural populations: excitatory or pyramidal ($e$) and inhibitory or interneurons($i$), and two thalamic populations: reticular nucleus($r$) and relay nuclei($s$). Previously, this model has been tuned to reproduce the electrical brain activity in different states: from arousal to slow-wave sleep and including the different stages of NREM sleep [29, 30]. In this NFT model, sensory stimulation can be introduced as spatial-localized perturbations on the noisy input neural population, obtaining an EEG-like signal as the model's output. From this signal, we can assess the efficacy of the stimulation paradigms and implement a closed-loop feedback mechanism that can fine-tune the stimuli delivery timing.

With the purpose of simulating a single-modality sensory stimulation, we searched for a one-channel impulse pattern that alters the dynamics of the entire cortex. We assessed the efficacy of different pulse shapes, delivered with different frequency, intensity, and periodicity, including a closed-loop algorithm that intended to phase-lock the stimuli with the SO. The stimulation goals were:

1. To increase the power at both frequency bands of interest (SO and SP), measured by the scalogram (wavelet spectrogram).

2. To increase the quantity of SO and SP events and detecting their occurrence on the time series.

3. To rise the co-occurrences between SO and SP events, above the level of co-occurrences obtained by chance.

4. To increment the occurrence of SP on the up-peak of the SO, measured by the delay time between the down-peak of the slow oscillation and the center of the coincident sleep spindle.

Our results propose a sensory stimulation protocol, maximizing the SP and SO co-occurrence associated with the memory consolidation process, for its use and verification in experimental setups of memory consolidation enhancement. Specifically, the SP and SO events' best co-occurrence is obtained when 50 ms decreasing ramp pulses were delivered at the zero

phases (sine reference) of the ongoing activity filtered at a specific central frequency on the SO band. Nevertheless, the behavioral effect of this stimulation protocol on memory performance should be evaluated with clinical experiments.

## Results

### Simulation of EEG recordings of NREM sleep

In EEG recordings, the classification as NREM sleep comes from the presence of SWS events on epochs of 30 seconds. The prominence of SO indicates the N3 stage, and the occurrence of SPs indicates the N2 stage [31]. Furthermore, the evidence suggests that SP nested in the up phase of the SO may play a role in memory consolidation [15]. A SWS model, including SOs and SPs, presents advantages to characterize the conditions promoting their co-occurrence. We used the neural-field theory model with thalamic and cortical populations, *eirs*-NFT [27], with the connections shown in Fig 1A to obtain simulated EEG-like recordings. The spatio-temporal propagation of the firing rate in each population (*e,i,r,s*) is used as input of the synaptic-dendritic dynamics of the soma voltage in other populations (Fig 1B). The connection strengths and the steady-state of the input population $\phi_n^{(0)}$ determines the behavior of the

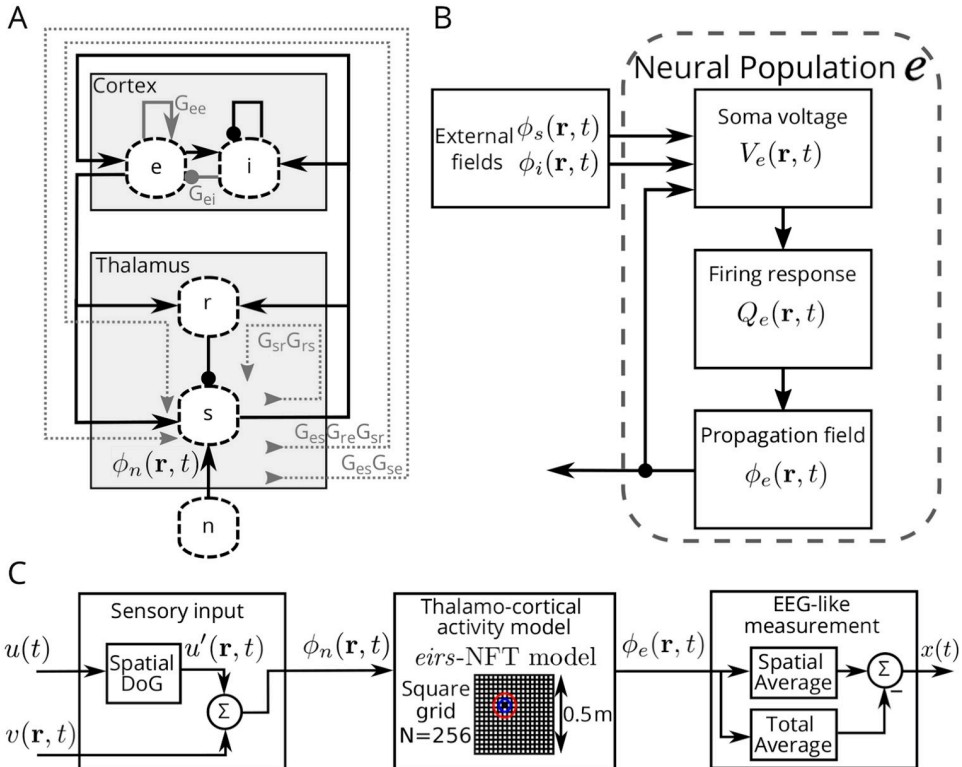

**Fig 1. Neural Field Theory model and additional modules.** (A) *eirs*-NFT model connections and loops. Arrow ended connections are excitatory, and dot ended connections are inhibitory. There are two intra-cortical loops with gains $G_{ee}$ and $G_{ei}$ related to the X-dimension (see Fig 2A), one intra-thalamic loop with gain $G_{sr} G_{rs}$ related to the Z-dimension, and the gains $G_{es} G_{se}$ and $G_{es} G_{re} G_{sr}$ are cortico-thalamic gain-loops related to the Y-dimension. (B) Diagram of the dynamics of a population of the Neural Field Theory. Particularly, the diagram shows as example the excitatory population *e* of the *eirs*-NFT model. (C) Complete simulation diagram including the modules of spatial coupling of the sensory stimulation input and the spatial averaging at the output. The grid in the cortico-thalamic activity block shows the nodes' spatial distribution. The location of stimuli is highlighted by the circles in color, representing the standard deviations of the Difference of Gaussians spatial filter.

model (see Eq (7) in the Materials and methods section) by changing the value of the connection gains $G_{ab}$, being $a$ the receiving population and $b$ the delivering one.

Our model includes a stage for sensory stimuli input (left block of Fig 1C), consisting on Gaussian white noise along all the spatial extension, $v(\mathbf{r},\mathbf{t})$, to which the stimuli signal $u'(\mathbf{r},\mathbf{t})$ is added when applied. The stimuli signal location comes from applying a Difference of Gaussians filter to the stimuli signal $u(t)$. A spatial filter acts as a receptive field, with excited and inhibited areas, on the relay nuclei of the thalamus. Finally, the output is the spatio-temporal activity of the excitatory population $\phi_e(\mathbf{r}, t)$, which is averaged in the spatial domain to generate a simulated EEG-like signal (right block of Fig 1C).

## Selection of model parameters for SWS stage

Previous works using the *eirs*-NFT model were specific for the N2 stage (only SP) [29], or for the N3 stage only (without SP) [30]. Then, our first goal was to tune the model parameters into a regime that exhibits both events.

In the *eirs*-NFT model, the sleep (and awake) stages can be represented as points in a XYZ space of model parameters, related to physiological connections. Fig 2A shows this space where the X-axis represents the strength of the intra-cortical connections, higher as sleep deepens. The Y-axis represents the internal thalamocortical connections that have a negative value during sleep. The more pronounced variations between sleep stages are in the Z-axis, representing the intra-thalamic connections [30]. In terms of the loop gains, the axes are defined as:

$$X = \frac{G_{ee}}{1 - G_{ei}}$$

$$Y = \frac{G_{es}G_{se} + G_{es}G_{re}G_{sr}}{(1 - G_{sr}G_{rs})(1 - G_{ei})}$$

$$Z = -G_{sr}G_{rs}\frac{\alpha\beta}{(\alpha + \beta)^2},$$

where $\alpha$ and $\beta$ are the rising and decaying rate of the synaptic-dendritic potential, respectively. Then, the position in the XYZ space depends on the model parameters and $\phi_n^{(0)}$.

Furthermore, Fig 2A shows excerpts of the cortical excitatory population activity for N2 and N3 stages, and we traced a trajectory between these two stages using linear interpolation on the connection strengths. On top of the cortical activity, the root mean square (RMS) value of the bands shows the higher presence of low-frequency SO activity (orange line) when the point is at a low Z-value. The SP activity (green dashed line) increases as the point moves towards more negative values on the Y-axis. Dominant spindle activity is appreciable using the N2-stage parameters indicated in [29]. The spindle activity decreases as the simulation point slides along the interpolated trajectory towards the N3 stage, which shows slow-wave sleep dynamics but not noticeable spindle activity.

We are interested in an intermediate state between N2 and N3. For this, we concentrated on having energy both in N2 and N3 frequency bands. We obtained this condition in the second half of the linear interpolation between N2 and N3. Within this segment, we arbitrarily selected the point at 2/3 on the N2 to N3 trajectory. Fig 2B shows the linearized model spectrums of N2 and N3 stages and three extra points in the interpolation trajectory, including the 2/3 position in the blue-line.

The increase of the steady-state input value ($\phi_n^{(0)}$) raises the model response in the Z-axis (Fig 2A), indicating lightness of NREM sleep [30, 32]. Besides, it also increases the power and

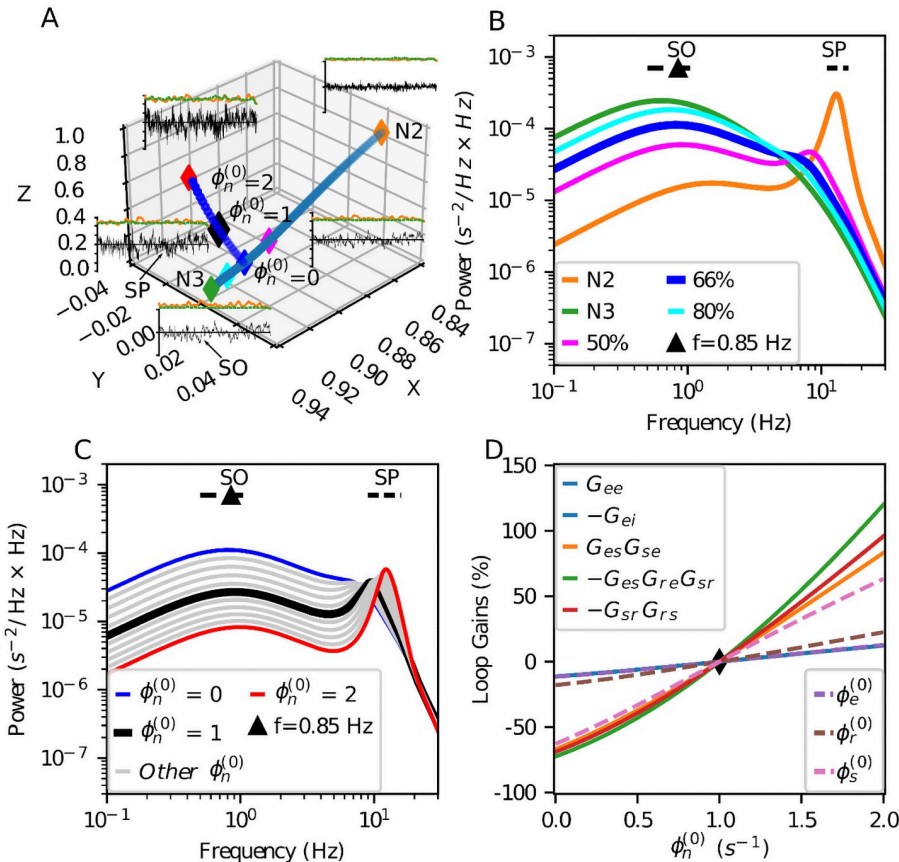

**Fig 2. Selection of baseline simulation parameters.** (A) Trajectories of parameters searching in the XYZ-space. Selected parameters point indicated with a black diamond; this point is at 2/3 of the interpolation trajectory with $\phi_n^{(0)} = 1$ s$^{-1}$. The change of the steady-state input $\phi_n^{(0)}$ does not affect the X-axis value. The excerpts duration is 20 seconds, and y-axis ticks indicate 0.01 s$^{-1}$ around the subtracted average cortical activity. The orange line represents the RMS value of SO band activity, and the green dashed line is the RMS value of SP band activity. (B) Dynamics of the model for parameters in the linear interpolation between N2 and N3 sleep stages with $\phi_n^{(0)} = 0$ s$^{-1}$. The selected point of parameters is closer to the N3 stage, keeping some additional energy in the SP band compared to the N3 published parameters [30]. (C) Dynamics of the model while changing $\phi_n^{(0)}$ in the selected point of simulation at the interpolation trajectory. (D) Relative changes with respect to $\phi_n^{(0)} = 1$ s$^{-1}$ of the loop gains and the neural population's steady-state response values for the same cases of panel C. The loop-gain with more change is $G_{es} G_{re} G_{sr}$. The change of $\phi_e^{(0)}$, $G_{ee}$, and $G_{ei}$ are similar and they are the less sensitive to the variation of $\phi_n^{(0)}$

frequency of the SP-band peak (Fig 2C), then a higher steady-state input value facilitates SP occurrence.

We selected $\phi_n^{(0)} = 1$ s$^{-1}$ as steady-state input in order to have similar peak amplitudes for frequency band lobes of both SWS events. The black line in Fig 2C shows the corresponding spectrum of the chosen simulation parameters. The peak frequency is close to the lower SP frequency with the selected parameters for the connections gains and the steady-state input (see Materials and methods).

The variation of the input, expected with the addition of stimuli signal, has a high effect in the relay nuclei activity $\phi_s^{(0)}$, and a less considerable effect in the reticular nucleus activity $\phi_r^{(0)}$, and the cortical activity $\phi_e^{(0)}$. Fig 2D displays the relative changes of the loop-gains and the steady-state of the neural populations for the changes of the steady-state input activity.

## Enhancement of SWS events on the *eirs*-NFT model

We searched for changes in the occurrence of the slow-wave sleep events of interest (SO and SP) when we added stimulation $u(t)$ to the noisy input propagation field $\phi_n(\mathbf{r}, t)$. Slow oscillations (SO) were defined to occur every time the signal, representing the cortical activity at the band 0.5–1.25 Hz, had a negative peak below -40 μV, and a peak-to-peak amplitude higher than 75 μV. We selected the higher amplitude oscillations at the SO band (see Materials and methods). On the other hand, sleep spindles (SP) were defined as bursts of oscillations in the 9–16 Hz band.

Fig 3A shows the simulated cortical activity for three different stimulation types: random (STIM-R), periodic (STIM-P), and closed-loop with a target-phase (STIM-CL). The top panel shows the simulated cortical activity for the SHAM condition, $b(t)$. STIM-R stimulation delivers stimuli pulses at random times from a Poisson stochastic process with a mean inter-stimulus time. With STIM-P, the pulses occur at a fixed inter-stimulus period. STIM-CL follows the ongoing activity at a narrow band delivering a stimulus every time the signal arrives at a target-phase (see below, section SWS events activity variation with closed-loop stimulation). Fig 3B shows the correspondent scalograms for each time series in Fig 3A, where $\delta$-band activity (0.5–4 Hz) –that includes SO-band (0.5–1.25 Hz)– is present through all the time. On the other hand, the SP-band activity (9–16 Hz) occurs at discrete periods, some of them with enough duration and amplitude to be classified as spindles.

Fig 3C focuses on a smaller time window when two SPs and one SO occur. The vertical marks in Fig 3A and 3C show the start time point of each stimulus pulse that, in this case, consists of decreasing ramps of 0.1 s duration and 40 a.u. of energy. The horizontal marks in Fig 3A and 3C represent detected SWS events. Their duration and peak-to-peak amplitude have the distributions shown in Fig 3E and 3F, respectively.

Using the time-averaged scalograms, we calculated the spectrums shown in Fig 3D. We defined the *power difference index*, $I^{(SO, SP)}$, as the area between the corresponding STIM condition and SHAM spectrums inside the frequency band of interest, normalized by the sum of the power of both cases. Then, $I^{(SO, SP)}$ quantifies the differences with respect to the SHAM case for the frequency bands of interest (see Materials and methods).

## Selection of the stimulus pulse characteristics with open-loop stimulation

We characterized the efficiency of different stimulation signals to enhance the occurrence of slow-wave events. In a first step, we tested stimulus pulse features with the open-loop stimulation (STIM-R and STIM-P). We looked at power differences at the frequency bands of interest and the number of occurrences of SWS events while changing the shape, the energy, and the duration of the stimulus pulse. In the following, we will evaluate each of these parameters' impact on the co-occurrence of SP and SOs.

**Changes on SWS events by pulse shape.**   With STIM-P and STIM-R, the shape, duration, and pulse energy change the temporal and frequency domain characteristics of $u(t)$. The stimulation frequency also modifies them in the case of STIM-P. We tested for the six shapes in Table 1, and shown in Fig 4A and 4B, with the same duration (0.1 seconds), keeping constant the stimulus pulse energy ($E_p$ = 40 a. u.) and the stimulation frequency (0.2 pulses per second). Fig 4C shows changes in the SO, and Fig 4D shows changes in the SPs. The power difference index (*x*-axis: $I^{(SO)}$, $I^{(SP)}$) and the number of occurrences (*y*-axis: $N_{SO}$/min., $N_{SP}$/min.) are plotted simultaneously for each type of SWS event. The decreasing ramp overpasses the results of all other shapes in both events and both measurements with significant differences (Welch's t-test, $p < 0.01$).

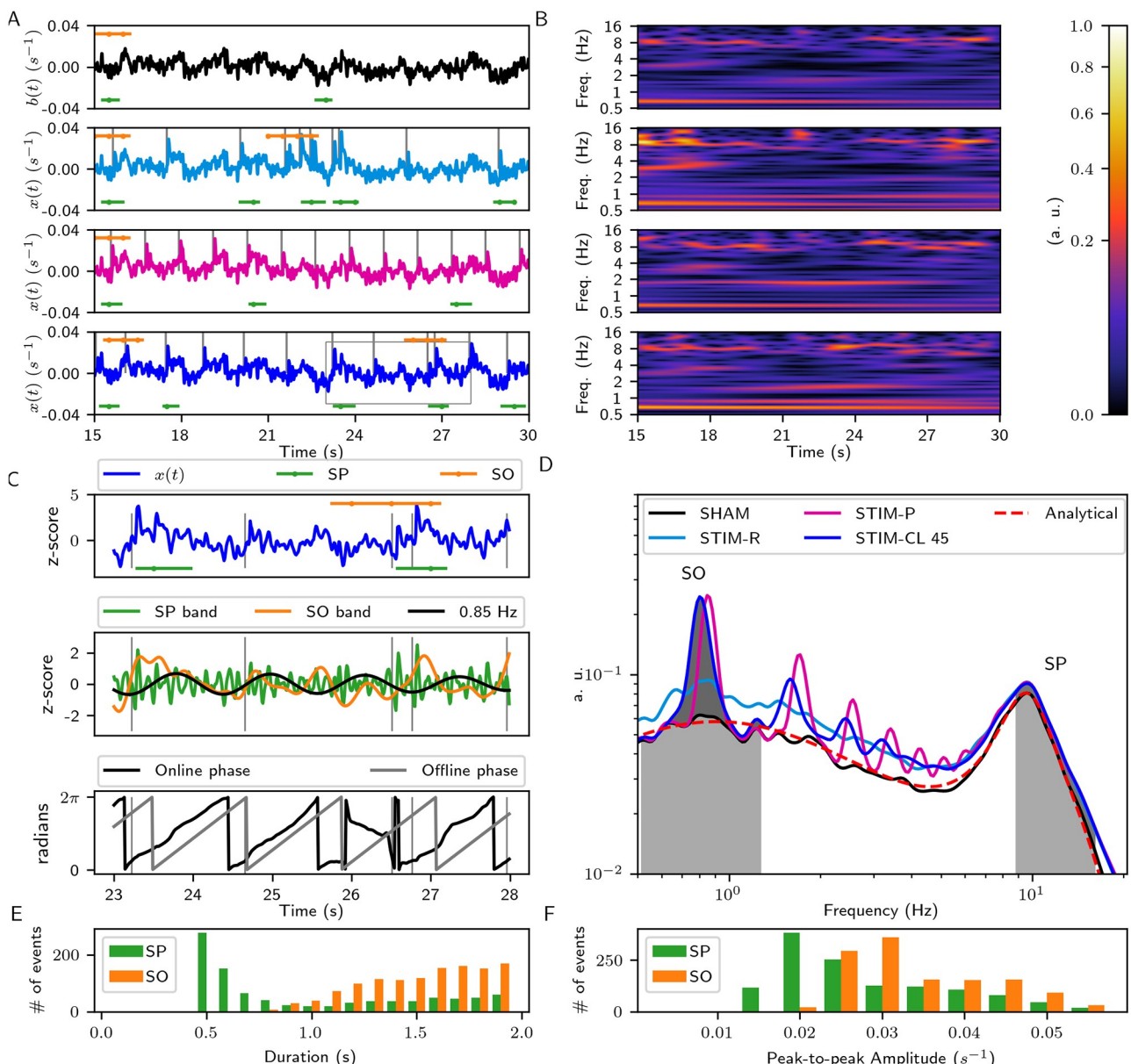

**Fig 3. External stimuli on the eirs-NFT model.** (A) Excerpts from the time series of the stimulation. From top to bottom, SHAM, STIM-R, STIM-P, and STIM-CL simulations. Vertical bars show stimuli delivery. The detected events above each time series correspond to SOs, and the detected SPs are below. The dots in the marks of detected events represent intervals of 0.5 seconds. The stimulation parameters are: pulse shape: decreasing ramp, pulse energy = 40 (a. u.), duration = 0.1 s, target-phase of closed-loop = 45 degrees, closed-loop (STIM-CL) and periodic (STIM-P) stimulation frequency = 0.85 Hz, random stimulation (STIM-R) times come from a Poisson distribution with $\lambda = 1/0.85$ s. (B) Wavelet scalograms of SHAM, STIM-R, STIM-P, and STIM-CL activity shown in (A). The increase of the SO-power is visible in the scalograms, and the increase of both SWS events occurrence is notable with their detection in the time domain. (C) Zoom of the box of STIM-CL time series. Top, z-score of $x(t)$, and the detected SWS events. Middle, signal filtered at the event-bands and at 0.85 Hz for phase detection in STIM-CL. Bottom, comparison of the online detected phase and the offline phase got by Hilbert transform. (D) Wavelet spectrums calculated by the time average of the scalograms in (B) and the scaled analytical spectrum. Note that the shadowed area between the curves STIM-CL and SHAM is the numerator in the calculation of $I^{(SO)}$ for STIM-CL (Eq (8)). (E) Distribution of the duration of detected events with the SHAM, STIM-R, and STIM-P stimulations. (F) Distribution of the peak-to-peak amplitude of the detected events with the SHAM, STIM-R, and STIM-P stimulations (events extracted from $x(t)$ which has firing rate units).

**Table 1. Stimulus pulse shapes.**

| Shape | Energy $E_{pulse}$ | Stim. Amplitude |
|---|---|---|
| Rectangular | $A^2 D$ | A = 20.00 |
| Gaussian | $0.3957 A^2 D$ | A = 31.79 |
| Rectangular trapezoid | $2A^2 D/3$ | A = 24.56 |
| Triangular | $A^2 D/3$ | A = 34.64 |
| Rising ramp | $A^2 D/3$ | A = 34.64 |
| Decreasing ramp | $A^2 D/3$ | A = 34.64 |

Pulse energy calculation and pulse amplitude to obtain an energy value of 40 (a. u.) for each shape with a duration of 0.1s.

The co-occurrence of *SP* and *SO* was evaluated using: (i) the conditional probability of counted co-occurrences given the number of detected spindles, $P(C|SP)$; and (ii) the probability of slow oscillations, $P(SO)$. In Fig 4E, a positive result means that $P(C|SP)$ is higher than $P(SO)$; in other words, that the probability of spindles co-occurring with SOs is higher than chance. Given this, both ramp shapes (decreasing and increasing) have positive results for the co-occurrence of events.

**Changes on SWS events by duration and energy of stimulus pulse.** Other stimulus pulse characteristics could modify the SWS events measurements. The pulse duration modifies the influence of stimulation on the frequency domain (narrower stimulus, wider spectral

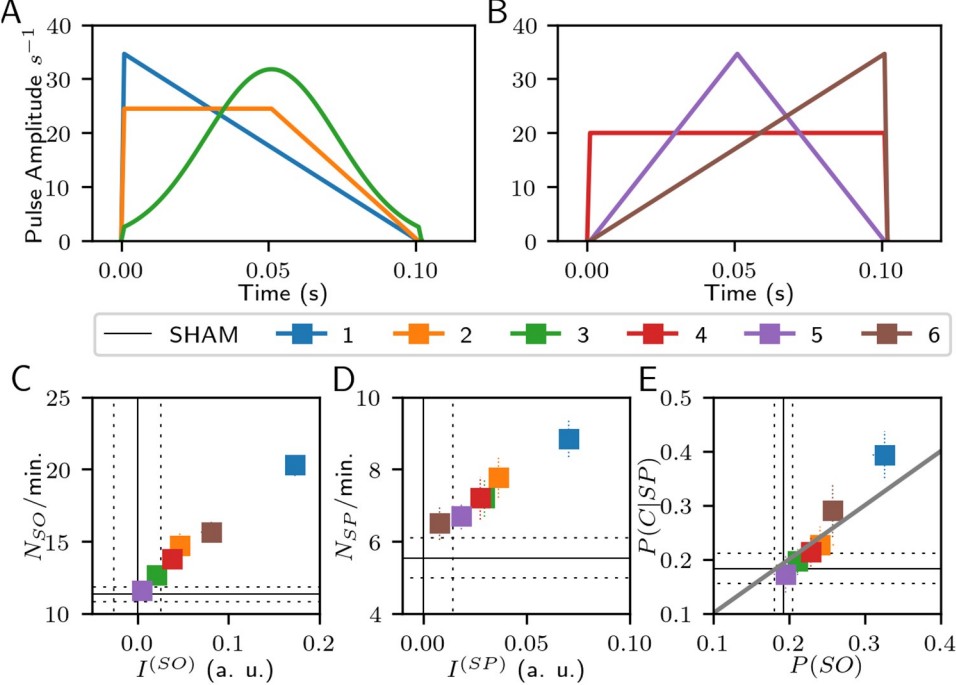

**Fig 4. Stimulation results changing the shape of the stimulus pulse with rhythmic stimulation.** (A) The tested pulse shapes are the decreasing ramp (1), rectangular trapezoid (2), Gaussian (3); (B) rectangular (4), triangular (5), and rising ramp (6). These panels show the pulse shapes with the same $E_p$. (C) Changes in SOs occurrence by stimulus shape. (D) Changes in SPs occurrence by stimulus shape. (E) Changes in the co-occurrence of both SWS events by stimulus shape. The identity-diagonal in (E) represents the chance to explain the temporal coincidences by the time percentage of SO occurrence.

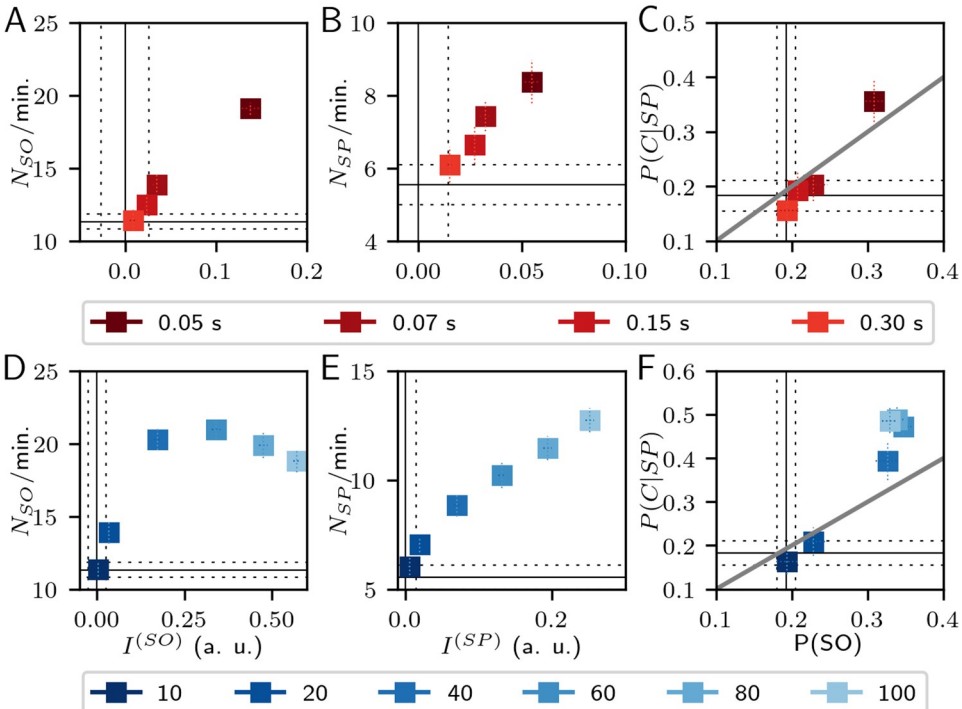

**Fig 5. Stimulation results changing one parameter of the stimulus pulse with rhythmic stimulation.** Top:(A) Changes in SOs by pulse duration. (B) Changes in SPs by pulse duration. (C) Changes in the co-occurrence of both SWS events by pulse duration. Bottom: (D) Changes in SOs by pulse energy. (E) Changes in SPs by pulse energy. (F) Changes in the co-occurrence of both SWS events by pulse energy. Solid black lines are the average values of SHAM condition, and dashed lines indicate one standard deviation. Each column shares the same horizontal axis labels. The identity-diagonal in (C), and (F) represents the chance to explain the temporal coincidences just by the time percentage of SO occurrence.

response), and changes in $I^{(SO)}$ and $I^{(SP)}$ are expected from rising the pulse energy. We tested for different pulse duration keeping constant the pulse energy (40 a. u., rectangular pulses in order to keep the amplitude along the pulse, and avoiding slope changes. Fig 5A–5C), and for various pulse energies keeping constant the duration (0.1 s, decreasing ramp pulses. Fig 5D–5F), both by changing the pulse amplitude. Among the tested duration values, the 0.05 s pulse displays better results for all measurements with statistical significance ($p < 0.01$), except in $N_{SP}$/min. vs. 0.075 s pulses ($p > 0.03$).

The minimum pulse energy was selected to overpass the input-noise power of 9.67 s$^{-2}$ per second in each node. $I^{(SO)}$ and $I^{(SP)}$ increase as the pulse energy rises with statistical differences at each increase step ($p < 0.01$), but $N_{SO}$/min. and $N_{SP}$/min. detected by stimulation with pulse energy of 100 (a. u.) are not different from the detected events with pulse energy of 80 (a. u.) ($p > 0.08$). These measurements are also not different between the minimum tested pulse energy and the SHAM condition ($p > 0.1$). Interestingly, Fig 5D shows a peak of $N_{SO}$/min. at 60 (a. u.), but without statistical difference from their neighboring pulse energies. The decrease of $N_{SO}$/min. with higher pulse energies arises from the impossibility of evoking more SO events without overlapping them. Regarding the co-occurrence of events, the rectangular pulse shape of 0.05 s, and the decreasing ramp pulses of 0.1 s with energies above 40 (a. u.) show positive results in Fig 5C and 5F, respectively.

The overall results of Fig 5 suggest that the best choice is the 0.05 s duration pulse, always considering that lower duration translates to higher amplitude in order to keep the same pulse

energy, which in turn must be enough to overpass the intrinsic activity. Nevertheless, optimal results are found without need to rise over a six-fold value of the intrinsic power per second.

When exploring the stimulation frequency using a decreasing ramp of 0.1 seconds, we found no statistical differences between frequencies in the 0.5–1.25 Hz range (see S2 Fig). Stimulation with pulses at random intervals (STIM-R) has very similar effects of periodic pulses (STIM-P), except for a higher $N_{SO}$/min. with STIM-P than STIM-R (see the first row of Fig 5 and the second row of S3 Fig). Nevertheless, the results keep the same order between the characteristics variation. Thus, the difference of effects in Fig 5 comes from the pulse characteristics and not from the stimulation periodicity.

## SWS events activity variation with closed-loop stimulation

Next, we asked whether the phase of the ongoing activity could influence the power and the number of slow-wave sleep events at the moment of the stimulus onset. To answer this, we implemented a closed-loop stimulation protocol. Here, the stimulus pulse is applied when the inverse-notch filtered signal arrives at a particular target phase (see Fig 6A and Materials and methods section). Our phase detector uses a fixed frequency of the ongoing activity, and 0.85 Hz has been used in other closed-loop implementation as the central frequency of a phase-locked loop (PLL) stage [33, 34].

We applied the decreasing ramp stimulation pulses of 0.1 s duration and $E_p$ = 40 (a. u.) in the three stimulation cases: STIM-P, STIM-R, and STIM-CL. STIM-CL was further divided into STIM-CL 0 (stimulus onset when the phase of slow oscillations arrives at 0 degrees; the upside cross-zero point), STIM-CL 45 (45 degrees) and STIM-CL 90 (90 degrees; up-peak of the slow oscillation).

The STIM-CL 0 case outperforms the other two closed-loop stimulations in all the measurements as shown in Fig 6B–6D, with statistical significance in $I^{(SP)}$ and $P(SO)$ ($p < 0.01$ Welch's independent t-test if both cases pass Shapiro, otherwise Wilcoxon test, see S1 and S2 Tables for all t-values and p-values). The STIM-CL 0 case is also statistically different from STIM-CL 90 for $I^{(SO)}$.

In relation to the open-loop stimulation cases, STIM-CL 0 outperforms STIM-P and STIM-R in $I^{(SO)}$ ($p < 0.01$ vs STIM-P), $I^{(SP)}$ ($p < 0.01$ vs STIM-R) and $N_{SP}$/min. Interestingly, STIM-P outperforms all other stimulations in $N_{SO}$/min. but is only statistically different with STIM-R. These results probably come from the zero chance in STIM-P of two pulses occurring with an inter-pulse interval less than the stimulation frequency period.

The co-occurrence of events shows positive results in Fig 6D for all stimulation cases. STIM-P and STIM-CL 0 are very similar in the $P(SO)$ versus $P(C|SP)$ plane, but this plot does not say anything about the timing of occurrence of these temporal coincidences.

## Timing of SWS events occurrence

A higher occurrence, or higher power of slow-wave events, may not be enough to enhance memory consolidation [35]. The relative timing of occurrence between SPs and SOs also play a role, with SPs being expected to be nested in the up-phase of a slow oscillation. Fig 7 shows histograms of the time delays between the SO's down-peak and the center of the SP from coincident events. Fig 7 also displays the average amplitude of the SOs, taken from $x(t)$, showing how the timing of the centers of the SPs are related to the down- and up-phase of SOs.

The histograms in Fig 7 include all coincident events from the total simulation time per stimulation condition. All stimulation conditions produce more SPs at the up-peak of SOs. Moreover, STIM-CL 90 condition is statistically different to the STIM-CL 0 and SHAM cases (Kolmogorv-Smirnov test, $p < 0.05$, see Table 2). SPs' timing distribution from STIM-CL 45 is

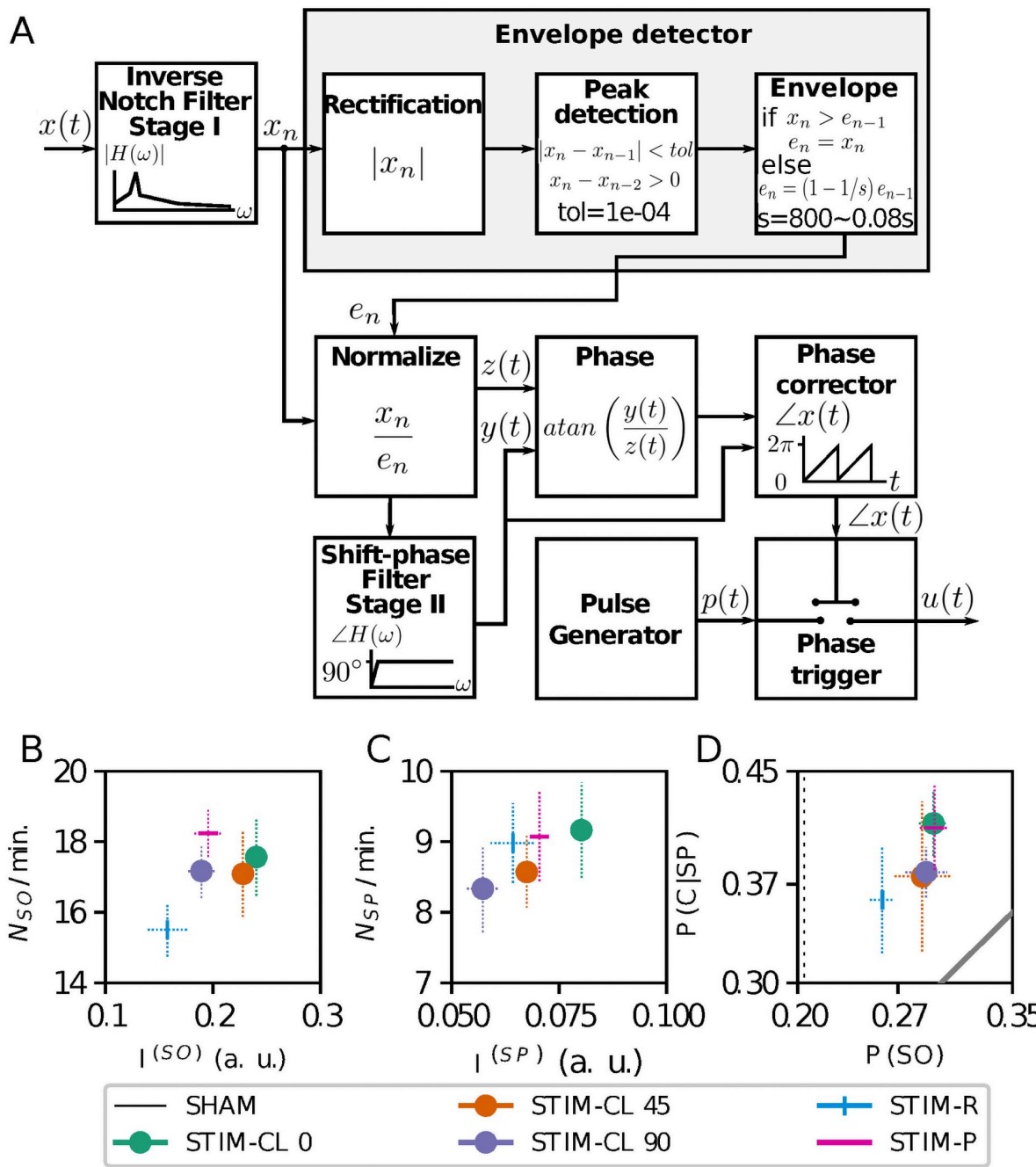

**Fig 6. Closed-loop stimulation driver and results with different stimulation type.** (A) Diagram of the closed-loop driver. The phase of a specific frequency from output signal $x(t)$ triggers the pulse generator when it arrives at the target-phase. At the bottom, changes in SWS events at a desired phase closed-loop stimulation, STIM-P, and STIM-R. (B) Changes in the SOs number of events and power index $I^{(SO)}$. (C) Changes in the SPs number of events and power index $I^{(SP)}$. (D) Changes in the co-occurrence of both events. The identity-diagonal represents the chance to explain the temporal coincidences just by the time percentage of SO occurrence.

also significantly different from the SHAM distribution ($p < 0.01$). Other significant differences ($p < 0.05$) are STIM-R vs. SHAM, and STIM-R vs. STIM-CL 0. The STIM-CL 45 and STIM-CL 90 show better SPs' timing, but the co-occurred events and the other measurements are higher with STIM-CL 0. Then, STIM-CL 0 is the best among the tested cases of stimulation.

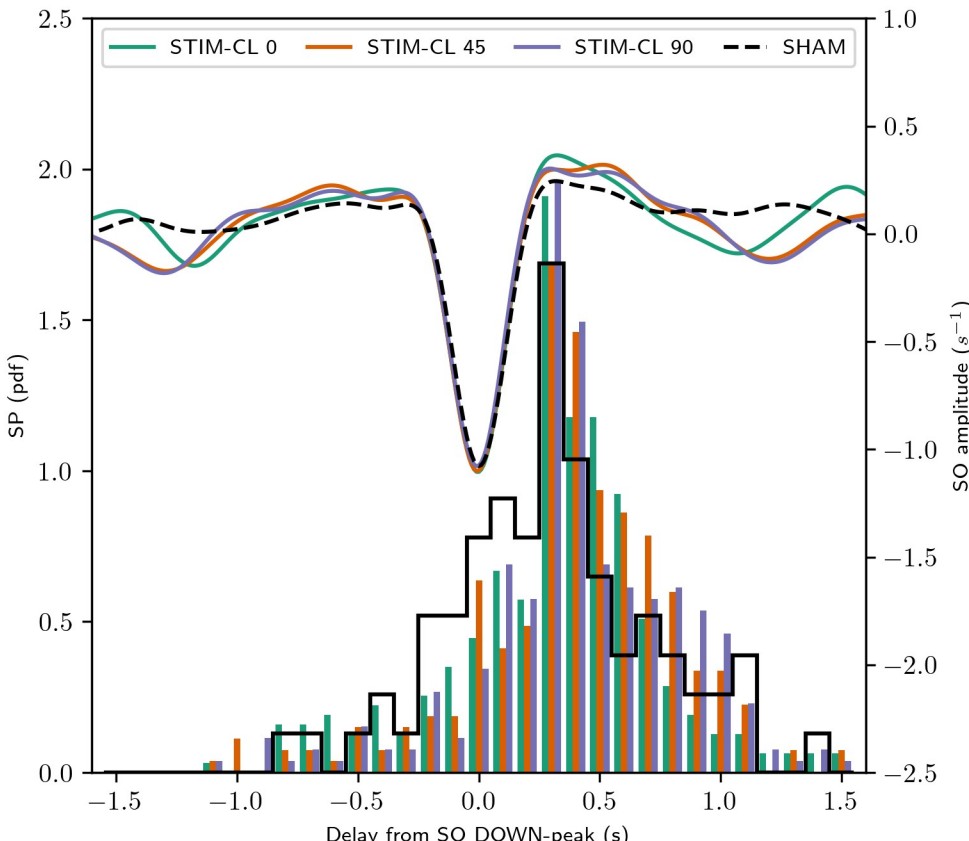

**Fig 7. SO down-peak locked results with different stimulation type.** Changes in the temporal occurrence of SWS events at the three closed-loop stimulation conditions. The histograms of spindles occurrences consider the entire simulation time per each stimulation type. The left vertical scale indicates the number of spindles with the center at the time delay from the SO down-peak indicated in the horizontal axis. The averaged time series at the top come from all detected SOs and all simulations using a decreasing ramp pulse, with pulse energy = 40 (a. u.), duration = 0.1 s, stimulation frequency, and central frequency for phase extraction = 0.85 Hz. For each stimulation condition, the simulation duration was fixed to 180/0.85 ($\sim$ 212) seconds producing the following number of pulses: STIM-CL0: 936; STIM-CL45: 815; STIM-CL90: 755.

Further, inter-stimuli interval distribution is different between the simulation cases (see S5 Fig). A stimulus modifies the ongoing activity, increasing the activity slope, even when the pulse is delivered nearer to the up-peak. Then, STIM-CL 45 and STIM-CL 90 delay the peak occurrence, thus needing more time to detect the same target phase again. Moreover, the

**Table 2. Statistical test results of spindles occurrence distribution.**

|               | SHAM | STIM-R  | STIM-P | STIM-CL 0 | STIM-CL 45 | STIM-CL 90 |
|---------------|------|---------|--------|-----------|------------|------------|
| Co-occurrences | 77   | 267     | 304    | 317       | 267        | 263        |
| SHAM          | -    | *0.011  | 0.116  | 0.173     | **0.008    | *0.015     |
| STIM-R        | -    | -       | 0.115  | *0.032    | 0.509      | 0.868      |
| STIM-P        | -    | -       | -      | 0.840     | 0.367      | 0.116      |
| STIM-CL 0     | -    | -       | -      | -         | 0.137      | *0.026     |
| STIM-CL 45    | -    | -       | -      | -         | -          | 0.638      |

Kolmogorov-Smirnov test p-values from searching for dissimilarity on the distribution of time-delay from the down-peak of the coincident slow oscillations and spindles. The mark '*' indicates $p < 0.05$, and '**' indicates $p < 0.01$.

distribution of delays of the SP centers in Fig 7 shows more presence of spindles after the SO up-peak with STIM-CL 45 and STIM-CL90, indicating that the phase of pulse delivery makes changes in the SO and SP timing.

## Discussion

In this study, we used a neural field model with thalamocortical populations to represent the SWS activity. We then explored the parameter space to maximize the occurrence of the SWS events (SOs and SPs), maximizing their frequency power and co-occurrence. In the open-loop, among different pulse shapes, we propose the decreasing ramp pulse, with a pulse duration of 50ms, as the one maximizing the SWS features. On the other hand, the closed-loop stimulation, with stimulus applied at zero degrees of the ongoing activity (filtered at a specific frequency), increases the power and the number of both SWS events, together with their co-occurrences. Here, we analyze in detail the advantages and limitations of our contribution.

### Selection of stimulation parameters

This research asks which stimulation pattern is effective in raising the power of slow-wave sleep events, consequently increasing the SP and SO occurrences. Moreover, we were also interested in maximizing the co-occurrence of SP with the up-peak of SO.

Our most significant result is that a pulse of 0.05 seconds with decreasing ramp shape maximizes the power and number of both SWS events. The features of the stimulation pulses delivered with STIM-P that result in higher $I^{(SO)}$, $I^{(SP)}$, $N_{SO}$/min., and $N_{SP}$/min. are also above the diagonal of chance in the plane $P(C|SP)$ vs $P(SO)$. It is interesting to note that in experimental acoustic stimulation studies, the pulse duration was 50 ms [3, 33, 36], a right choice following our results for pulse duration.

The pulse energy must be higher than the intrinsic activity. Following our results, the preferred energy value is 60 (a. u.). However, the translation of this energy value to a physical measurement of sensory stimulation is difficult. Thus, the useful recommendation is to use an energy value around six times the intrinsic EEG activity power per second.

To achieve SPs' precise timing at the up-peak of SOs, STIM-CL 45 and STIM-CL 90 conditions significantly differ in the timing distribution compared with the SHAM condition. The evoked response's effect (see S1 Fig) shifts the signal's peak, mainly because the activity slope—which was about to reach zero—increases up to the following maximum. On the other hand, the stimulus generates a burst activity at the thalamus that could last enough to cause cortical SPs. These responses on both brain regions could justify the change of the timing distribution. However, the STIM-CL 0 condition provides a better result in power and number of events than STIM-P except in $N_{SO}$/min. as shown in Fig 6B, and with the higher number of co-occurrences indicated at Table 2. Morevoer, the STIM-CL 0 condition actually shortens the inter-stimuli interval.

**The pulse shape and its onset amplitude reshape the SWS events response.** The shape of the stimulation pulses had small relevance in the previous neuro-stimulation works [37]. One probable cause is the technical difficulty of performing the stimulation following a particular shape-wave with precision in short duration times. For example, the triangular shape in Fig 4A and the Gaussian shape in Fig 4B may be indistinguishable from each other if the stimulation device has a small sampling rate. However, it is essential to consider that the mean firing rate from all neural populations does not represent the modulated amplitude of the sensory pulse linearly.

In any case, and regarless of the amplitude modulation, the pulse shapes in this work may represent variations in the firing response from different populations; each shape producing a

different outcome from the thalamocortical system (see S4 Fig). Then, the rectangular shape relates to constant firing response. The rectangular trapezoid and the decreasing ramp are responses with short-term adaptation [38]. The Gaussian and triangular relate to a set of heterogeneous populations with different time-lag for the response's start. Finally, the rising ramp, less naturalistic, could be seen as an exaggeration of the sequential response of time-lag populations. The presented shape of firing response includes the external sensory noise, but it excludes the variation by other brain internal sources, which is assumed the input noise $\phi_n(\mathbf{r}, t)$.

At the cellular level, the work of Pyragas et al. [39] demonstrates that using Pontryagin's maximum principle (a control theory formulation), a bipolar bang-off-bang waveform is optimal for the entrainment of a neuron to a specific frequency. This waveform type is not suitable for sensory stimulation, because a sensory stimulus affects excitatory and inhibitory neuronal populations at the same time [40]. With basis on the receptive fields of the visual perception system and with the assumption of the stimulus independence from the constant variance Gaussian noise input [41], we used a difference of Gaussians as a spatial filter of the simulated sensory input.

Different pulse shapes could generate benefits for stimulating during sleep, like soft slopes for avoiding abrupt sensory inputs of high prediction error and contrast-dependent temporal dynamics. A sudden stimulus is detrimental in sleep by evoking non-desired arousal. However, the results for inducing SPs and SOs were better when each pulse began with a high amplitude (see also S3 Table). The decreasing ramp shapes produce more beneficial and notable changes for both SWS events. In the same direction, small duration pulses with higher initial amplitude for keeping the same pulse energy show a higher increase in the power of both SWS events bands and a higher $N_{SP}$/min. in Fig 5B. Furthermore, the rising ramp (time-mirrored version of the decreasing ramp) results in Fig 4D have the lowest performance of the tested shapes. These results suggest high relevance of the onset amplitude and the intrinsic phase of the stimulus pulse. Secondary effects of these stimuli, such as non-desired arousal, will have to be experimentally determined.

## Results-driven by the model dynamic linear and non-linear effects

Mean-field models of the brain activity relay in two assumptions: The average population dynamics are similar to one neuron dynamics, and the average is enough statistic to characterize the population activity [27, 42]. Further, the output activity of a mean-field neural population depends on the dynamics (see Eq (1) in Materials and methods) and statistics of the incoming signals (Eq 4). In addition to the $u(t)$ dynamics, the input activity's statistics could play a role on the output dynamics of the model.

As in previous works of the Neural Field Theory [28–30, 32], we use Gaussian white noise as input, which provides the linear impulse response of the model. The noisy input activity represents the net contribution from other sources as the basal ganglia, brainstem, and peripheral nervous system [30]. At the minute scale, variations from the average input value come from these region's activity rather than modifications to the connection's strengths. In our simulations, we used a permanent average value for the noisy input.

We superimpose a pulse train in this approach, modifying the input mean value, and its variance. Still, the steady-state results in Fig 2C do not predict all the dynamic effects produced by adding a pulse train at the model's input.

The most noticeable difference from the expected results with the steady-state approximated response appears when the tested stimulation parameter is the pulse energy. Fig 5D and 5E show that with the rise of pulses energy, there is an increase in the power of both SWS

events bands. The increase of $I^{(SO)}$ is opposite to the expected result from Fig 2C when the steady-state input rises. This increase in power does not come only from the periodicity of stimulation. The random stimulation also increases the event's bands' power similarly (see S3 Fig). Moreover, in conjunction with results in S2 Fig, a specific stimulation frequency may not be too relevant for enhancing the power of both SWS events. It is also important to note that there are no significant differences in the results between STIM-R simulations.

Another possible way to change the input signal's statistics, and consequently its steady-state value, is by modifying their spatial characteristics. The coupling with sensory regions could matter to get results and predictions with sensory stimulation. However, the nodes' position in our model does not have a particular relation with a cortical region. Results from harmonics studies [43, 44] make it possible to assign a physiologically-related topography in future works. Consequently, the results reported here are not related to a particular sensory type; they are general in how the increase of activity in a delimited region could modify the whole system's output.

More work is needed to consider non-linear effects that surpass the expected response using the linear approximation to explain the obtained results with stimulation. The use of modern control techniques [45], and the integration of information theory measurements [46] could provide a better analytical approximation of the results than traditional linear systems analysis.

## Relation to similar computational works

The work of Schellenberger et al. [10] tries to reproduce in a neural-mass model the experimental results from Ngo et al. [3] with closed-loop stimulation. They recover the experimental results using two external calcium-dependent currents and an anomalous rectifier current responsible for the waxing and waning structure of the spindle oscillations. The authors employed the same neural populations of *eirs*-NFT with one additional connection, the auto-connection of the thalamic reticular nucleus, and two additional noise inputs to the cortical populations. Similar to our approach, the sensory stimulation (80 ms square pulses with amplitude of 70 s$^{-1}$) is added to the noise input of the relay nuclei. The simulation results show high similarity with the SO band's experimental ones on amplitude and timing, but differences in the SP band activity's timing.

Wei et al. [26, 47, 48] tested several ways of closed-loop stimulation, including the protocol of Ngo et al. [3], in a cellular level model. Their results showed that the correct spatio-temporal localization of stimuli could facilitate the replay of a cortical sequence associated with the stimulation site, pointing to the need for peaks nesting for strengthening the neural connections. For this purpose, they propose stimuli application at the end of the SO's down-state, near the start of the SO up-state (0° by sine reference), almost matching with our stimulation target-phase proposal.

## Relation with experimental works and further work

Non-invasive methods of neural system stimulation operate at large spatial scales, and they can be by direct application of electrical or magnetic fields or by the use of sensory stimuli to modify neural activity [37, 49]. The non-invasive techniques are under active-research because of the potential clinical applications in humans. Several works [17, 18] tried to enhance the SWS activity to achieve memory improvements. There were applications of olfactory [50, 51], auditory [3, 33], and tactile stimuli [52] with similar or even better results than direct stimulation [53–59].

Following the discoveries from previous works of Ngo et al., the application of stimulus at a precise phase of the ongoing cortical activity [3] produces better behavioral results than rhythmic stimulation [36, 60]. Another result of the same authors shows that the application of more than two consecutive pulses does not improve physiological outcomes [61]. These results suggest the presence of mechanisms that prevent an over-driving of SO activity, which could be present in our case given a similar increase of $I^{(SO)}$ but lesser $N_{SO}$/min. with STIM-R (see S1 Fig). A recent work [35] shows that this closed-loop stimulation pattern does not always enhance memory consolidation even with the increase of the power of the SWS events. Furthermore, based on the stimulation protocol of Ngo et al. [3], the recent work [62] identifies the phase at which the amplitude of SO increases, also observing an increase in the SP quantity and amplitude. They encountered the SO up-peak, or 90˚, as optimal timing for stimuli application.

Phase detection for closed-loop stimulation depends on the selected SO frequency band. Pulse stimulation at the SO rising slope increments its frequency power [3, 33], which was also observed in our results for the three closed-loop conditions studied (see $I^{(SO)}$). However, precisely determining the signal phase to stimulate either at zero-cross, or peak, highly depends on the selected SO frequency band.

Our results show that the shape, energy, and application phase of the stimulation pulses modifies the power and occurrence of SWS events. On the framework of EEG, these stimuli parameters are often neglected in sensory event-related potential (ERP) analysis, given the confounds with attention in awake experimental studies [63] and the incomplete knowledge about the codification pathways of sensory information. Despite this, the stimuli used in this work represent incoming firing responses to the thalamus, not the sensory stimuli properly. However, these stimulation characteristics are under research for neural implants, and direct stimulation [64]. In contrast to in-vivo works, one limitation of our work is the assumption of a steady-state of the sleeping brain during the entire simulation time, ignoring the changes known to occur within the same sleep stage. Finally, the selected stimulation parameters need experimental verification to correlate with memory task performance.

## Conclusion

The neural field model can reproduce SWS activity with the occurrence of slow oscillations and sleep spindles. Rhythmic, random, and closed-loop stimulation was applied using this model to enhance the SWS events. Closed-loop stimulation, with stimulus applied at zero degrees as target phase of the ongoing activity filtered at a specific frequency, increases the power and the number of both SWS events, together with their co-occurrences. Also, closed-loop stimulation at other target-phases changes spindle timing with respect to the down-peak of slow oscillations. Our results indicate that the number of slow oscillations decreases when the stimulation pulse energy surpasses more than six times the background noise power per second. On the other hand, increasing the stimulus energy increases the SPs' power and their number of occurrences. The stimulus that promotes a higher occurrence of both events is the decreasing ramp shape with a duration of 50 ms applied at target-phase zero of closed-loop stimulation.

## Materials and methods

### Large-scale brain model: Neural field theory

Neural field theory represents the the activity of a whole population of neurons as its mean firing rate. Spatial propagation can then be calculated to obtain the spatio-temporal evolution of neuronal activity [27].

Thalamocortical interactions can be modeled using Neural Field Theory, where both cortical and thalamic regions are each represented by a pair of excitatory and inhibitory neural populations. The populations relate to known structures. In the cortex, the excitatory population $e$ corresponds to the pyramidal neurons, while inhibitory population $i$ represents the inter-neurons. In the thalamus, the excitatory population represents the relay nucleus $s$ of incoming sensory information. Similarly, the thalamic inhibitory population represents the reticular nucleus $r$. Fig 1A (see Results) schematizes the connections between the four populations in the *eirs*-NFT model.

The dynamics of the mean soma voltage, $V_a$, of the neural population $a$, with incoming activity from another population $b$ (with $a$, $b$ being elements of the set of neural populations **p** = $\{e, i, r, s\}$) is represented as (see also Fig 1B)

$$\frac{1}{\alpha\beta}\frac{d^2 V_a(\mathbf{r}, t)}{dt^2} + \left(\frac{1}{\alpha} + \frac{1}{\beta}\right)\frac{dV_a(\mathbf{r}, t)}{dt} + V_a(\mathbf{r}, t) = \sum_{\mathbf{p}} v_{ab}\phi_b(\mathbf{r}, t - \tau_{ab}),\tag{1}$$

where $\alpha$ is the inverse of the decaying time of the impulse response, and $\beta$ is the inverse of the rising time of the impulse response. Both parameters shape the post-synaptic-dendritic response. $v_{ab}$ is the strength of the connection from population $b$ to $a$, and its sign determines whether the connection is excitatory (positive connection strength) or inhibitory (negative connection strength). $\tau_{ab}$ is the propagation delay between the populations $a$ and $b$. Then, the model uses a sigmoid function to convert the soma voltage into the firing rate population response, $Q_a$, as follows

$$Q_a(\mathbf{r}, t) = S[V_a(\mathbf{r}, t)] = \frac{Q_{\max}}{1 + \exp\left(-(V_a(\mathbf{r}, t) - \theta)/\sigma_\rho\right)},\tag{2}$$

where $Q_{\max}$ is the maximum firing response, and the sigmoid function approximates the normal cumulative probability distribution of the firing threshold voltage with mean $\theta$ and standard deviation $\sigma$ where $\sigma_\rho = \sigma\sqrt{3}/\pi$.

The spatial propagation of the firing response only applies to the excitatory population $e$, the one with axons long enough for making it relevant [28]. It is expressed as

$$\frac{1}{\gamma_e^2}\frac{\partial^2 \phi_e(\mathbf{r}, t)}{\partial t^2} + \frac{2}{\gamma_e}\frac{\partial \phi_e(\mathbf{r}, t)}{\partial t} + \phi_e(\mathbf{r}, t) - r_e^2\nabla^2\phi_e(\mathbf{r}, t) = Q_e(\mathbf{r}, t),\tag{3}$$

where $\gamma_e = v_e/r_e$ is the temporal damping coefficient for pulses propagating at a velocity $v_e$ through axons with a characteristic range $r_e$. For the other populations, $\phi_a(\mathbf{r}, t) = Q_a(\mathbf{r}, t)$.

The expressions Eqs (1)–(3) govern the dynamics of each neural population. Their parameters are physiologically related values [28] and are listed in Table 3. Note that the subscripts of the parameters follow the order: *'input to'—'output from'*. Fig 1B (see Results) represents the connections inside a population and relationship between the variables.

Table 3 presents the parameters used in our simulations of SWS activity on the *eirs*-NFT model.

**Linear transfer function and loop-gains.**   To calculate the spectrum of the system's linear approximation with diverse sets of connections strengths, we used the steady-state solutions of the Eqs (1) and (3). The steady-state solution of the corticothalamic model comes from setting all temporal and spatial derivatives to zero. Defining $\phi_e^{(0)}$ and $\phi_n^{(0)}$ as the steady-state value of the cortical neural population and the input, respectively, the steady-state

**Table 3. Parameters selected for SWS with spindles.**

| Symbol | Value | Unit |
|---|---|---|
| $\alpha$ | 45 | s$^{-1}$ |
| $\beta$ | 186 | s$^{-1}$ |
| $\tau_0$ | 0.085 | s |
| $r_e$ | 0.086 | m |
| $\gamma$ | 116 | s$^{-1}$ |
| $Q_{max}$ | 340 | s$^{-1}$ |
| $\sigma_\rho$ | 0.0038 | V |
| $\theta$ | 0.01292 | V |
| $v_{ee}$ | 5.54 | mVs$^{-1}$ |
| $v_{ei}$ | -5.65 | mVs$^{-1}$ |
| $v_{es}$ | 1.53 | mVs$^{-1}$ |
| $v_{re}$ | 0.286 | mVs$^{-1}$ |
| $v_{rs}$ | 1.12 | mVs$^{-1}$ |
| $v_{se}$ | 2.69 | mVs$^{-1}$ |
| $v_{sr}$ | -1.73 | mVs$^{-1}$ |
| $v_{sn}$ | 9.22 | mVs$^{-1}$ |
| $\overline{\phi_n} = \phi_n^{(0)}$ | 1 | s$^{-1}$ |
| SD($\phi_n$) | 3.11 | s$^{-1}$ |

Parameters of the model used in all the simulations. The first three parameters correspond to the synaptic-dendritic and soma-voltage dynamics Eq (1)). $Q_{max}$, $\sigma_\rho$, and $\theta$ are parameters related to the activation function of the firing response (Eq (2)). $r_e$ and $\gamma$ are parameters of the spatiotemporal propagation (Eq (3)). The following eight parameters are the connection strengths between the neural populations. Finally, the two last parameters are the mean and standard deviation (SD) of the Gaussian white noise input.

solution [29] is

$$S^{-1}[\phi_e^{(0)}] - (v_{ee} + v_{ei})\phi_e^{(0)} = v_{es}S[v_{se}\phi_e^{(0)} + v_{sr}S[v_{re}\phi_e^{(0)} + (v_{rs}/v_{es})(S^{-1}[\phi_e^{(0)}] - (v_{ee} + v_{ei})\phi_e^{(0)})] + v_{sn}\phi_n^{(0)}]. \quad (4)$$

We searched the steady-state solutions that accomplish Eq (4) using the function *fsolve* of Matlab with an initial value for $\phi_e^{(0)} = 10$ s$^{-1}$.

The linear approximation of the sigmoid function in Eq (2) uses its Taylor's serie expansion

$$Q_a(\mathbf{r}, t) = Q_a^{(0)} + \rho_a(V_a(\mathbf{r}, t) - V_a^{(0)}) + O^2, \quad (5)$$

where $\rho_a$ is the slope of the sigmoid function evaluated at $V_a^{(0)}$, the steady-state soma voltage of the population $a$. Note from Eq (3) that, in the steady-state, $\phi_a^{(0)} = Q_a^{(0)}$ and we can treat each variable of the model as the perturbation from their steady-state value, reducing the above expression to

$$\phi_a(\mathbf{r}, t) = Q_a(\mathbf{r}, t) = \rho_a V_a(\mathbf{r}, t). \quad (6)$$

Taking the Fourier transform of Eqs (1), (3) and (6), and rearranging and eliminating $\phi_r$ and $\phi_s$, leads to the system's spectrum with output $\phi_e$, and input $\phi_n$ [29, 30].

$$\frac{\phi_e(\mathbf{k}, \omega)}{\phi_n(\mathbf{k}, \omega)} = \frac{L^2 G_{es} G_{sn} \exp(i\omega t_0/2)}{(1 - LG_{ei})(1 - L^2 G_{sr} G_{rs})(\delta - \mathbf{k}^2 r_e^2)}, \tag{7}$$

$$\delta = \left(1 - \frac{i\omega}{\gamma_e}\right)^2 - \frac{1}{1 - LG_{ei}}\left[\frac{G_{ee}L + (G_{es}G_{se}L^2 + G_{es}G_{re}G_{sr}L^3)\exp(i\omega t_0)}{1 - L^2 G_{sr} G_{rs}}\right],$$

where $G_{ab} = \rho_a \nu_{ab}$ is the gain of the connection $ab$, and $L$ is an abbreviation for $L(\omega)$, the reciprocal of the Fourier transform of the operators applied to $V_a(\mathbf{r}, t)$ in Eq (1) and expressed as

$$L(\omega) = \left(\left(1 - \frac{i\omega}{\alpha}\right)\left(1 - \frac{i\omega}{\beta}\right)\right)^{-1}.$$

The linear approximation in Eq (7) uses a reduction of the model parameters with the same incoming connection strengths for the excitatory and inhibitory populations of the cortex, i. e. the connection strength $\nu_{ee} = \nu_{ie}$, $\nu_{ei} = \nu_{ii}$, and $\nu_{es} = \nu_{is}$. The gray arrows in Fig 1A represents the gain loops in Eq (7); $G_{ei}$ becomes an additional loop gain with the reduction of parameters. Further, the non-zero propagation delays considered in Eq (7) only correspond to the ones between the corticothalamic and thalamocortical populations, $\tau_{es} = \tau_{se} = \tau_{re} = t_0/2$.

**Spatial coupling of stimuli input.** The *eirs*-NFT model uses spatio-temporal variables, but our stimuli input is a single-channel time series. To give the stimuli a spatial representation, the signal goes through the sensory input module (the block at the left of Fig 1C in Results). We formed a receptive field using a Difference of Gaussians (DoG) spatial kernel:

$$u'(\mathbf{r}, t) = \frac{u(t)}{\sqrt{2\pi}\sigma_E}\exp\left(\frac{(\mathbf{r} - \mu_\mathbf{r})^2}{\sigma_E^2}\right) - \frac{u(t)}{\sqrt{2\pi}\sigma_I}\exp\left(\frac{(\mathbf{r} - \mu_\mathbf{r})^2}{\sigma_I^2}\right).$$

The spatial filter is centered at an interior node $\mu_\mathbf{r}(x = 7, y = 7)$ of the square sheet of 256 nodes with positive standard deviation $\sigma_E = 1$ and negative standard deviation $\sigma_I = 2$. The chosen center and standard deviations values avoid the grid boundaries.

Before entering the thalamocortical model, we added Gaussian white noise, $\nu(\mathbf{r}, t)$, forming the potential field of the $n$th population, $\phi_n(\mathbf{r}, t)$.

**Simulation procedure.** We solved the model dynamics in Eqs (1)–(3) using the Euler's integration method with a time step $h = 10^{-4}$ seconds, and a uniform sheet of $N = 256$ equidistant nodes in a square cortex of 0.5 meters per side using periodic boundary conditions (toroidal boundary conditions). The model was implemented on Matlab, and it is available on https://github.com/mjescobar/SWSNeuralField. The simulations were performed on a computer running Ubuntu Server 18.04 with a double Xeon E5–2630 v4 processor.

Each simulation lasts 910 seconds. We performed five simulations for every tested stimulation pattern $u(t)$, each with a different random number generator seed, giving a total of 4550 seconds per stimulation case. The stimulation started after the first 5 seconds and stopped before the last 5 seconds in open loop stimulation. The initial conditions of neural populations variables come from a previous 6-second simulation with initial conditions equal to zero, and without stimulation.

We stored the output cortical activity $\phi_e(\mathbf{r}, t)$), and the input signal $\phi_n(\mathbf{r}, t)$ with a sampling frequency of 100 samples per second.

We obtained an EEG-like signal, $x(t)$, spatially averaging the output activity for all time points, and then we subtracted its temporal mean value

$$x(t) = \frac{1}{N}\sum_{\mathbf{r}}\phi_e(\mathbf{r}, t) - \bar{\phi}_e(\mathbf{r}, t),$$

where $\mathbf{r}$ is the grid position, $N$ the number of nodes, and $\bar{\phi}_e(\mathbf{r}, t)$ the average of the cortical activity (see Fig 1C, module at the right).

## Event detection in the time domain

We detected the slow-wave sleep events (slow oscillations and sleep spindles) from the output signal of the complete model. In the following, we refer as $x(t)$ to the output signal of the stimulation cases (STIM), while $b(t)$ represents the output signal of the non-stimulation cases or SHAM condition.

**Detection of slow-oscillations events.** As our EEG-like signal, $x(t)$, is not in voltage, we cannot use the classical method to detect SOs in EEG recordings. The classical method establishes SOs every time the signal has a negative peak below -40 μV, and a peak-to-peak amplitude higher to 75 μV. As an alternative, we searched for events with time-lengths between 0.8 and 2 seconds (frequency: 0.5–1.25 Hz) [3], and we kept the higher amplitude oscillations.

The detection of SOs is based on the search for high amplitude oscillations in a delta-band sub-region. First, we filtered the signal in the SO frequency band (0.5–1.25 Hz, Chebyshev type I filter, fourth-order, $10^{-6}$ dB allowed ripple, zero-phase lag). Second, we used the z-score of the filtered signal for searching the zero-crossing points. Later, we applied a threshold to the filtered signal, and we searched for negative peaks below the arbitrary low value of $-10^{-6}$ s$^{-1}$. Using the zero-crossing points, we calculated the peak-to-peak amplitude for the single oscillations that underpass the negative peak threshold. Finally, slow oscillations are confirmed if the peak-to-peak amplitude is higher than 1.25 times the baseline peak-to-peak amplitude average.

**Detection of sleep spindles.** Spindles are bursts of oscillations in the 9–16 Hz band. To detect them, we normalized each simulated output $x(t)$ by the mean and variance of their corresponding $b(t)$ signal. The results are single-channel time series expressed as z-scored EEG registers for each simulation. Then, we computed the RMS value of the z-score $x(t)$ filtered in the SPs frequency band (9–16 Hz, Chebyshev type I filter, fourth-order, $10^{-6}$ dB allowed ripple, zero-phase lag) and applied an amplitude threshold for the automatic detection of SPs [23, 65]. The RMS value calculated at each sample uses a 0.2 s window length, and we also smoothed it with a Hamming window of the same time length [66]. SPs are then detected every time the signal exceeds a detection threshold of 1.25 standard deviations of the RMS value of the filtered baseline signal.

**Counting of co-occurrent events.** We defined co-occurrence of events every time SOs and SPs overlap by at least 250ms. The counting process of co-occurrences uses tagged events to avoid duplication. We also extracted the time percentage of occurrence of SOs, SPs, and co-occurrences. From here, we use the identifier $N_{SO}$ for the number of slow oscillations, $N_{SP}$ for spindles, and $N_C$ for co-occurrence of events.

## Power increases in the event's frequency bands

We define the scalogram power difference index ($I^{(j)}$), which relates the power of the oscillations in the $j$th frequency band as Eq (8)

$$I^{(j)} = \frac{\sum_{s \in s_j} \sum_{t \in T} |S_x(s,t)| - |S_b(s,t)|}{\sum_{s \in s_j} \sum_{t \in T} |S_x(s,t)| + |S_b(s,t)|}, \quad j = SO, SP , \tag{8}$$

where $S_x(s, t)$ is the STIM register scalogram, $S_b(s, t)$ is the baseline or SHAM register scalogram. $s_{SO}$ comprises the scales inside the 0.5–1.25 Hz spectrum related to slow oscillations and the $s_{SP}$ the range of scales correspondent to the frequency range of the sleep spindles of 9–16 Hz. $T$ is the total simulation time.

We used the Morlet wavelet in the scalogram computation. This wavelet shape is a Gaussian-modulated complex sinusoid with template $\Psi(t) = A \exp(-t^2/(2\sigma^2)) \exp(j\omega_0 t)$. The mother wavelet has central frequency $\omega_0 = 15$, using scaling factors, that result in a frequency resolution of 300 scales related to a starting frequency at 0.1 and ending at 30 Hz. The scales are equally distanced in a logarithmic range [67].

## Closed-loop phase-lock driver

The closed-loop stimulation requires the on-line detection of instantaneous phase of the ongoing activity, to apply stimuli at the target phase. For this, we filtered $x(t)$ with an IIR inverse-notch filter for $\omega_0 = 2\pi f_0 h$ with poles $z = \{0.9999\exp(\pm j\omega_0), 0\}$, and zeros $z = \{-1, -1, 1\}$, considering a scaling factor of $10^{-4}$. In addition, we extracted the envelope of the filtered signal at each sample to normalize itself. The envelope detector (see Fig 6A in Results) has three stages: (i) We detected the peaks of the rectified signal using its absolute value. (ii) We exponentially decreased the peak value, with a decay rate of 12.5 s$^{-1}$, until the detection of a new peak or the reaching of the limit of a sample counter. (iii) We normalized the filtered signal with the current envelope value at each sample.

On the normalized signal, $z(t)$, an additional IIR filter gets its $\pi/2$ shifted-phase version $y(t)$. This shift-phase filter has the poles $z = \{0.9999\exp(\pm j\omega_0), 0.999\}$, and zeros $z = \{-0.5 \exp(\pm 0.495\pi), -0.9\}$. The filter scaling factor is the half quotient of the sum of the denominator coefficients over the sum of the initial numerator coefficients.

The calculated atan($y(t)/z(t)$) gets the phase in the range (-$\pi/2$, $\pi/2$). We translated the phase to the range (0, 2$\pi$) or (0,360) using a cross-zero detector applied to the shift-phased signal.

The stimulus pulse generator is triggered when the phase arrives inside a hysteresis window of $\varepsilon = \pm 9$ degrees ($\pm 5\%$) around the desired target-phase. The trigger waits for the phase 1.9 radians or 342 degrees to be reached before resetting for the next pulse application.

We applied the stimuli at a specific target-phase of a particular frequency in the closed-loop condition. The signal goes through a narrow band-pass filter and we detected the instantaneous phase with the arc-tangent with an online $\pi/2$-shifted phase filtered version of the narrow band signal. We normalized the amplitude of the signal before the arc-tangent by the envelope amplitude. The envelope value came from a peak detector, where the peak value decreases exponentially with a decay rate of 12.5 s$^{-1}$ until the detection of a new peak.

## Statistical tests

We tested for normality of each set of results for each stimulation pattern with the Shapiro-Wilk test.

To compare between pairs of stimulation patterns following a normal distribution, we used the Welch's t-test. On the contrary, when at least one pattern does not follow a normal distribution, we used the Wilcoxon signed-rank test. We used the Kolmogorov-Smirnoff test for the similarity between the distributions of time delays between the down-peak of the slow oscillation and the center of the coincident spindle.

All the statistical analysis were implemented using the library scipy.stats 1.5.2 in Python 3.7.

## Supporting information

The supporting information contains results by other stimulation frequencies, random stimulation results not plotted in Figs 4 and 5, tables with values from statistical tests, and plots of the obtained results versus the onset amplitude of the stimulation pulses.

**S1 Fig. Evoked response potential.** Average activity of the cortical activity from simulations with different stimulus shapes. The subtracted baseline is selected from -450 ms to the stimuli onset. The stimulus onset is marked with the bold vertical line and the stimulus turnoff is marked with the dashed line. Shapes: (A) decreasing ramp (1), rectangular trapezoid (2), Gaussian (3); (B) rectangular (4), triangular (5), and rising ramp (6).
(PDF)

**S2 Fig. Power and events changes by different stimulation frequency.** (A) Changes by stimulation frequency on SO measurements. (B) Changes by stimulation frequency on spindles measurements. (C) Changes by stimulation frequency on probabilities of coincident events. (D) Changes by different mean $\lambda$ of random stimulation on SO measurements. (E) Changes by different mean $\lambda$ of random stimulation on spindles measurements. (F) Changes by different mean $\lambda$ of random stimulation on probabilities of coincident events.
(PDF)

**S3 Fig. Power and events changes by random stimulation.** Top:(A) Changes by shape in SO. (B) Changes by shape in spindles. (C) Changes in the probability of co-occurrence by shape. (D) Changes by pulse duration in SO. (E) Changes by pulse duration in spindles. (F) Changes by pulse duration in the probability of co-occurrence. (G) Changes by pulse energy in SO. (H) Changes by pulse energy in spindles. (I) Changes by pulse energy in the probability of co-occurrence.
(PDF)

**S4 Fig. Correlation of SWS events changes with the initial amplitude of the stimulation pulses.** (A) Onset amplitude of stimulus vs $I^{(SO)}$. (B) Onset amplitude of stimulus vs slow oscillations. (C) Onset amplitude of stimulus vs probability of co-occurrence of events. (D) Onset amplitude of stimulus vs $I^{(SP)}$. (E) Onset amplitude of stimulus vs spindles. (F) Onset amplitude of stimulus vs conditional probability of co-occurrence respect to the occurrence of spindles.
(PDF)

**S5 Fig. Inter-stimuli intervals.** Histograms of time between delivered pulses in each condition: (A) STIM-R, (B) STIM-CL 0, (C) STIM-CL 45, and (D) STIM-CL 90. The bar of STIM-P is plotted as reference of the central frequency 0.85 Hz, the value of the probability distribution is higher than 5. The mode of the STIM-R interpulse is below 0.5 seconds. The statistical mode on the inter-stimuli increase as the target-phase of the closed loop increases. The statistical mode for STIM-CL 45 is the nearest to the rhythmic interval (STIM-P).
(PDF)

**S1 Table. Welch's t-test of slow oscillations measurements.** The t-values and p-values for changes in power and occurrence of slow oscillation events by different stimulation case. The symbol '\*' indicates $p < 0.01$, and 'w' indicates that one stimulation case doesn't accomplish the Shapiro test for normality.
(PDF)

**S2 Table. Welch's t-test of sleep spindles measurements.** The t-values and p-values for changes in power and occurrence of sleep spindles events by different stimulation case. The symbol '\*' indicates $p < 0.01$, and 'w' indicates that one stimulation case doesn't accomplish the Shapiro test for normality.
(PDF)

**S3 Table. Welch's t-test for *P*(*SO*) and *P*(*C*|*SP*).** The t-values and p-values for changes in the probability of occurrence of slow oscillations and co-occurrences of events by different stimulation phase. Welch's t-test applied after the confirmation of normality by Shapiro test. The '\*' indicates p-values <0.01. STIM-CL 0 number of spindles did not pass the normality test, and it neither did not pass the test with the SHAM condition, but is the highest value in the plot.
(PDF)

**S1 Video. Populations activity of the *eirs*-NFT model.** Video showing the activity of the *eirs*-NFT model populations with periodic stimulation (STIM-P) applied at 0.85 Hz using decreasing ramp pulses of 0.1 seconds duration and 40 (a. u.) pulse energy. The upper left panel shows the patch activity of the cortical excitatory population $\phi_e(\mathbf{r}, t)$, and the bottom panel shows the input activity $\phi_n(\mathbf{r}, t)$. The panels on the right show the time series of the spatial average subtracting the spatio-temporal mean of the population's propagation field. The shown populations start from the top with *e*, *r*, *s*, ending with *n* and the stimuli markers (in orange). Available on: Video link.
(MP4)

## Author Contributions

**Conceptualization:** Patricio Orio, María-José Escobar.

**Formal analysis:** Felipe A. Torres.

**Funding acquisition:** Patricio Orio, María-José Escobar.

**Investigation:** Felipe A. Torres.

**Methodology:** Felipe A. Torres.

**Project administration:** María-José Escobar.

**Resources:** María-José Escobar.

**Software:** Felipe A. Torres.

**Supervision:** Patricio Orio, María-José Escobar.

**Validation:** Patricio Orio, María-José Escobar.

**Visualization:** Felipe A. Torres.

**Writing – original draft:** Felipe A. Torres.

**Writing – review & editing:** Patricio Orio, María-José Escobar.

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
