## [Decision Letter · Decision Letter 0]

5 Mar 2021

Dear Dr Escobar,

Thank you very much for submitting your manuscript "Selection of stimulus parameters for enhancing slow wave sleep events with a Neural-field theory thalamocortical computational model" for consideration at PLOS Computational Biology.

As with all papers reviewed by the journal, your manuscript was reviewed by members of the editorial board and by several independent reviewers. The paper was overall well received, but some issues need to be addressed. In light of the reviews (below this email), we would like to invite the resubmission of a significantly-revised version that takes into account the reviewers' comments.

Please make sure to submit the code and the necessary data to reproduce the results, so that the editorial team and the reviewers can look at it. Even though the policy requires the data and code to be available before publication, I am sure that you will understand that being it part of the paper, it is fair to have it evaluated as well, also in order to allow us to suggest improvements to the presentation, ultimately increasing the impact and uptake of your work. At this review stage the code can be shared in a private repository, to be moved to a public one after acceptance. 

We cannot make any decision about publication until we have seen the revised manuscript and your response to the reviewers' comments. Your revised manuscript is also likely to be sent to reviewers for further evaluation.

Sincerely,

Daniele Marinazzo

Deputy Editor

PLOS Computational Biology

Daniele Marinazzo

Deputy Editor

PLOS Computational Biology

Reviewer's Responses to Questions

**Comments to the Authors: **

Reviewer #1: This paper uses an established corticothalamic neural field model to analyze the effects of various stimulation protocols on the occurrence of spindle (SP) and slow-oscillation (S) events, including the effects of stimulus timing relative to the SO. A key aim is to optimize stimulation to raise SO and SP power for potential applications to enhancement of memory consolidation.

This work uncovers optimal regimes for stimulation and makes predictions that can be tested experimentally. It can thus be expected to contribute significantly to the advancement of the field of brain stimulation for memory enhancement. 

Overall, the paper is very strong and suitable for publication in PLoS-CB so long as the authors consider the points below.

My comments are:

1) The authors should clarify a little more explicitly in the Introduction that their working hypothesis is that SOs and SPs enhance memory, rather than simply having a common cause with memory enhancement.

2) line 115: what are the units of phi_n^(0)?

3) table 1 and throughout the MS: physical units should be written in roman typeface, not italic. What is "std"?

4) p6: It might be worth mentioning that the SO and SP events are "evoked responses" - i.e., use this terminology because this is a large field and may attract interest from that community. This might be good to have as a keyword.

5) Table 2 needs a caption.

6) Fig 3 caption - "trapeze" should be "trapezium"

7) line 221: I'm not sure what was intended by the word "overpasses". Should it be "outperforms" or something similar? This word appears later in the MS too - e.g., line 558 where it perhaps should read "The trigger waits for the phase 1.9 or 342 degrees to be reached before ..."

8) line 294: should "notorious" be something like "notable". I'm sure that "notorious" is not what the authors intended.

9) line 415. The notation with b \\epsilon b will probably be a bit confusing to some neuroscientists. Perhaps just avoid it with a slight rewriting. Maybe say b \\epsilon p and mention the meaning of the epsilon symbol and state what elements are in the set p.

10) Eq 1 - first term on left has two sets of arguments (r,t).

11) the subscript max in Eq 2 and elsewhere should be in roman font - it's a mathematical function. Likewise atan on line 563

12) line 435 - "sub-indexes" should perhaps be subscripts

13) line 476 - write as 10^{-4} (i.e., with a superscript) rather than in computer notation 1e-4; likewise on line 506 and elsewhere

14) 488 - first argument of phi_e should be r

15) line 534 - please add a few words of description of the Morlet wavelet.

16 ) I suggest replacing "aleatory" by "random" throughout the paper - the former word does exist in English, but is exceedingly rare, despite its similarity to the Spanish word for "random" (I see that the authors are from a Spanish-speaking country).

17) In some places the paper could have been made clearer by using the present tense for the present work and past tense for past work - the authors use the reverse convention for some reason.

18) Title: The model is not a "computational" model. Computational methods are used to analyze it, but it is not formulated in terms of brain computations.

19) Fig 4D. Are we seeing a leveling off here because it is impossible to squeeze in more than about 20 SOs/min without them overlapping.

20) The caption of Fig 6 needs to refer explicitly to the curves at top. 

21) Supporting Information needs at least a brief preamble to explain what materials are included. 

22) It is much easier for the referees if the figures are placed where they are used in the MS, rather than at the end. If the journal requires them to be at the end they can be put there too or moved there upon acceptance.

Reviewer #2: This paper gives a comprehensive description of the authors’ implementation, parameter selection, numerical simulation and data analysis of their computational model, namely eirs-NFT model, based on the Neural Field Theory (NFT). The paper suggests that a decreasing ramp of 50 ms duration is the most effective amongst six stimulus shapes, and that the best effectiveness is achieved when the stimulus impulse is delivered in a closed-loop configuration targeting at the zero phase of the ongoing slow oscillation event.

Although the main contribution of the paper is its suggestion on the design of stimulus in real-life sleep experiments, the event detection methods are non-canonical from an experimentalist’s perspective. In "Event detection in the time domain", the authors acknowledged this limitation, and showed their endeavour in reducing any chance of false detection for frequency drifts or amplitude variability. However, it would still be interesting to know what is the interval of amplitudes (even with arbitrary units) detected for slow-oscillation detections. For spindles their detection is more straightforward. A graph summarising the events detected would be ideal. Even though it may not be accurately comparable to real-life results, it could offer more convincing evidence that their non-canonical method managed to detect events as expected.

The paper’s second major contribution is its deployment of the eirs-NFT model in the particular sleep stage. However, despite a brief mention in "Introduction" and a detailed recipe in "Materials and methods/Large-scale brain model: Neural Field Theory", a description of the model is missing from "Results/Selection of model parameters for SWS stage", making the XYZ space on page 4 and in Fig 1A and the chosen values of model parameters in Table 1 decontextualised and difficult to understand unless a reader is very familiar with previous works [24, 25] by other authors. The novelty of this work could thus be undervalued due to a lack of appreciation of the model and its parameters. At least the authors could give some verbal description in the text or in Table 1 to give readers (those unfamiliar with the model, or even the theory) some intuition why these parameters and the axes XYZ are important in the computational model, and how they are related to real-life experiments.

Miscellaneous on formatting:

Last line in Fig 1 caption on page 6: the subscriptions of G_{ee} and G_{ei} should be italic

Fig 4: One can probably guess the horizontal axes of A, B, C share the same labels as D, E, F, but it was not an easy guess for me because the scales are different. Or I guessed wrongly.

Line 273 on page 10: ‘Tab 3’ is inconsistent with previous notations.

Line 315 on page 11: missing space before ‘In our simulations, ...’

Line 415 & 435 on page 13 & 14: missing space before ‘7B’ 

Line 571 on page 18: obvious

Reviewer #3: The authors examined how SOs and SPs are enhanced by external inputs by using a mean-field model. By systematically modifying stimulation parameters, the authors have provided insight into how to optimise a close-loop stimulation protocol for memory consolidation during NREM sleep.

Major issues:

Although the authors have addressed a crucial issue in this field, the neurobiological significance of this study is not clear without supportive evidence to confirm the authors’ predictions.

Minor issues:

Introduction

The citations do not reflect this field well. I would suggest to cite papers/reviews written by Steriade, Buzsaki, and other pioneers in order to provide an appropriate context to readers.

Results

Because the Methods section comes later, provide the technical definition of SOs and SPs briefly. 

Figure 1A requires more explanation. For example, scales are missing for the example EEG traces. It is not clear when SPs appear.

In Fig. 3, what are the biological justification of all these pulses? Note that if this mean-field model assumes the auditory pathway, there are multiple auditory nuclei before signals reach the medial geniculate body. Thus, it is hard to believe that any sensory (acoustic) stimuli will appear as any of these. I would suggest that the authors should address how noises affect all measures at least.

In Fig. 4, only rectangular pulses were applied. How about other pulses?

In Table 3, the bottom left (or top right) half is redundant.

In Figure 6 and Table 3, although the outcomes of the close-loop stimulation are intriguing, the authors did not provide any explanations why these are the case. Further analysis will help to better understand (1) why the 0 deg stimulation induced more SO/SP couplings and (2) why 45/90 deg stimulation changed the SP distribution. In addition, what are impacts of this shift on the thalamocortical circuit dynamics?

Discussion

Before discussing technical issues, it would be better to summarize key observations first.

Neurobiological significant and limitations of this study must be discussed if any.

Reviewer #4: SUMMARY

=======

The article by Torres et al is a comprehensive investigation into the promotion of physiological sleep signatures using rhythmic sensory stimulation, within the framework of the corticothalamic neural field model of Robinson et al. The authors began with the reduced three-dimensional ‘XYZ’ parameter space of the linearized, spatially uniform steady state version of the model. Linear interpolating between previously identified N2 and N3 points in the XYZ space identified a point in parameter space showing both slow oscillation (SO) and sleep spindle (SP) events (peaks in the linear power spectrum). The authors then moved to the full nonlinear model, consisting of ‘alpha kernel’ synaptic response functions, sigmoidal wave-to-pulse transfer functions, a variety of input waveforms entering the thalamic relay population, and spatiotemporal wave propagation across the cortical sheet (here simply a 2D grid) following a damped wave equation. The spatial dimension is then collapsed down by averaging over space, yielding a scalar time series that is the principal object of study for the rest of the paper. The key result is that co-occurrence of SO and SP events was maximized by 50ms decreasing ramp pulses, delivered at the zero phase of filtered ongoing activity. 

RECOMMENDATION

Rhythmic sensory and electromagnetic stimulation for promoting sleep is an emerging area of study that has received considerable attention in recent years, both from academic science and commercial biotech. Work in this area has however been almost entirely atheoretical, outside of generic signal processing. As such, the in-depth physiologically-based investigation presented in this submission is a valuable, and fairly original, contribution to both the computational neuroscience and sleep physiology literature. 

I am therefore happy to recommend the paper for publication in PLoS Computational Biology, provided the authors can address the following issues: 

REVISIONS

=======

Major points 

--------------

Analytic vs. numerical power spectrum:

A central tenet of the study methodology is the selection of model parameters to study using the linear algebraic power spectrum model, and the further investigation of that point in parameter space with numerical simulations in the full PDE model. This is a very powerful general approach. However I think it is important to demonstrate correspondence between the analytic and numerical power spectra, for the selected parameters, by producing spectra with each method, and overlaying on the same graph. 

Relatedly: the power spectrum in Figure 2D seems very noisy. Indeed this is unsurprising, as the legend description indicates this is computed from only 15 seconds of data, out of the 910 seconds of simulation. Presumably, the effect of interest - enhancement at SO and SP frequency bands - would also appear, more strongly, in a power spectrum computed over several minutes, rather than 15 seconds. 

I would suggest to put both of these (analytic vs. empirical spectra, and a longer-run simulation) in a new Figure 2, and increment the figure numbers for Figure 2 onwards in the current text. But I leave it to the authors to decide how best to address this issue. 

Minor points

--------------

Implementation details:

Apart from the usage of Matlab fsolve for solution of Eqn. 4, there is not, so far as I have seen, a description of the model implementation in the methods. What programming language? Operating system? 

Reproducible code:

The questions in the opening comment boxes indicate that ‘all data will be made available’. But there are no further details on this. Presumably this is actually referring to code, which is the main outcome of this purely numerical simulation study. Please clarify what will be placed in the public domain, and where (github repository? Figshare? Supplementary file? )

Boundary conditions:

Please give some detail on the boundary conditions used for the numerical simulations. The Robinson model has been studied on planar, toroidal, and spherical domains. The relationship between the spatial damping rate and boundary conditions is a major emphasis in this work (e.g. Robinson et al. 1997), albeit with the general conclusion that it is not hugely important. 

Movie:

Given that activity is simulated across the spatial patch, it would be a nice touch to add a movie to the supplementary information, showing activity in the patch. 

Figure 2B:

Please reverse the y axis ordering for figure 2B. It is confusing having frequencies descending with increasing Y in a time-frequency plot. 

Figure text locations:

To whomever it concerns: it is very frustrating to have figures at the end of a text, with legends in the main text. Strongly recommend just put the figures in-line in the future in for-review manuscripts.

**Have all data underlying the figures and results presented in the manuscript been provided?**

Reviewer #1: None

Reviewer #2: None

Reviewer #3: **No: **The authors wrote, "All the data will be available after acceptance."

Reviewer #4: **No: **The data availability response says "All the data will be available after acceptance." More importantly: this is a simulation study. Data is not relevant per se; but the code is. There should be information on where the code will be provided. I have included this as a comment to be addressed.

PLOS authors have the option to publish the peer review history of their article (what does this mean?). If published, this will include your full peer review and any attached files.

Reviewer #1: No

Reviewer #2: No

Reviewer #3: No

Reviewer #4: **Yes: **John David Griffiths
---

## [Decision Letter · Decision Letter 1]

12 May 2021

Dear Dr Escobar,

Thank you very much for submitting your manuscript "Selection of stimulus parameters for enhancing slow wave sleep events with a Neural-field theory thalamocortical model" for consideration at PLOS Computational Biology. This version was significantly improved and overall appreciated. We will accept this manuscript for publication, but we're giving you the chance to modify or clarify the units on panel 3F.

Reviewer 2 also suggested a reference, you might certainly want to refer to it if you think it's relevant for the readers of this paper.

Sincerely,

Daniele Marinazzo

Deputy Editor

PLOS Computational Biology

[LINK]

Reviewer's Responses to Questions

**Comments to the Authors:**

Reviewer #1: The authors have dealt thoroughly with my previous comments and the paper is now suitable for publication.

I do want to stress that realistic predictions such as these are highly biologically relevant - they will provoke quantitative tests by experimentalists. It is certainly not necessary for the authors to do these experiments themselves - which would be a very unusual situation in more mature branches of science and engineering, where the need for specialization of theorists and experimentalists has been recognized for centuries.

Reviewer #2: We appreciate the authors' responses and changes. Two follow-up minor issues:

1. Why the unit of the amplitude in the new panel 3F is s^{-1}? Apparently this is due to the choice of arbitrary units, but this might be misleading to many readers.

2. This recent experimental paper (Navarrete et al 2020)[https://doi.org/10.1093/sleep/zsz315] might be interesting to the authors, as they were seemingly suggested the same timing for delivering stimulus during sleep.

Reviewer #3: The authors have addressed my initial concerns thoroughly. As a result, the revised manuscript has been improved significantly. Although the neurobiological significance of this elegant study is still an open question, I strongly believe that the current version is suitable for a publication.

Reviewer #4: The authors have addressed all my concerns and recommendations comprehensively. No further comments. Great paper, looking forward to seeing it published.

**Have the authors made all data and (if applicable) computational code underlying the findings in their manuscript fully available?**

Reviewer #1: Yes

Reviewer #2: Yes

Reviewer #3: Yes

Reviewer #4: Yes

PLOS authors have the option to publish the peer review history of their article (what does this mean?). If published, this will include your full peer review and any attached files.

Reviewer #1: No

Reviewer #2: No

Reviewer #3: No

Reviewer #4: **Yes: **John D Griffiths

Figure Files:

Data Requirements:

Reproducibility:

References:

---

## [Editor Report · Decision Letter 2]

28 May 2021

Dear Dr Escobar,

We are pleased to inform you that your manuscript 'Selection of stimulus parameters for enhancing slow wave sleep events with a Neural-field theory thalamocortical model' has been provisionally accepted for publication in PLOS Computational Biology.

Best regards,

Daniele Marinazzo

Deputy Editor

PLOS Computational Biology

Daniele Marinazzo

Deputy Editor

PLOS Computational Biology

---

## [Editor Report · Acceptance letter]

2 Jul 2021

PCOMPBIOL-D-21-00190R2 

Selection of stimulus parameters for enhancing slow wave sleep events with a Neural-field theory thalamocortical model

Dear Dr Escobar,

I am pleased to inform you that your manuscript has been formally accepted for publication in PLOS Computational Biology. Your manuscript is now with our production department and you will be notified of the publication date in due course.

With kind regards,

Katalin Szabo
